# Advances in Exosomes as Diagnostic and Therapeutic Biomarkers for Gynaecological Malignancies

**DOI:** 10.3390/cancers14194743

**Published:** 2022-09-28

**Authors:** Mengdan Miao, Yifei Miao, Yanping Zhu, Junnan Wang, Huaijun Zhou

**Affiliations:** 1Department of Gynaecology, Nanjing Drum Tower Hospital, Clinical College of Nanjing Medical University, Nanjing 210008, China; 2Department of Gynaecology, Nanjing Drum Tower Hospital, Nanjing University, Nanjing 210008, China; 3Department of Pharmacy, Nanjing University of Chinese Medicine, Nanjing 210023, China

**Keywords:** exosomes, advances, gynaecological malignancies, diagnostic markers, drug delivery

## Abstract

**Simple Summary:**

The three major gynaecological cancers are ovarian cancer, endometrial cancer, and cervical cancer, which endanger women’s health worldwide. Significant progress has been made in the study of exosomes, which have been proven to be an important form of intercellular communication, as well as an important carrier for the uptake, transport, and release of cargo. Exosomes may also be promising diagnostic or prognostic markers for gynaecologic malignancies, which may improve the level of treatment of gynaecologic malignancies. This article reviews the latest research progress and systematic knowledge of exosomes in gynaecological malignant tumours in recent years, in order to provide a new perspective for the treatment of gynaecological tumours and promote the clinical application of exosomes in gynaecological malignancies.

**Abstract:**

Background: Exosomes are extracellular vesicles that can be released by practically all types of cells. They have a diameter of 30–150 nm. Exosomes control the exchange of materials and information between cells. This function is based on its special cargo-carrying and transporting functions, which can load a variety of useful components and guarantee their preservation. Recently, exosomes have been confirmed to play a significant role in the pathogenesis, diagnosis, treatment, and prognosis of gynaecological malignancies. Particularly, participation in liquid biopsy was studied extensively in gynaecological cancer, which holds the advantages of noninvasiveness and individualization. Literature Review: This article reviews the latest research progress of exosomes in gynaecological malignancies and discusses the involvement of humoral and cell-derived exosomes in the pathogenesis, progression, metastasis, drug resistance and treatment of ovarian cancer, cervical cancer, and endometrial cancer. Advances in the clinical application of exosomes in diagnostic technology, drug delivery, and overcoming tumour resistance are also presented. Conclusion: Exosomes are potentially diagnostic and prognostic biomarkers in gynaecological malignancies, and also provide new directions for the treatment of gynaecological tumours, showing great clinical potential.

## 1. Introduction

Exosomes are membrane-coated particles that range in size from 30–150 nm and can transport several types of cargo, including proteins, lipids, genetic material, and others. Exosomes carry out intercellular communication and have an impact on recipient cells’ functionality. This reveals excellent research prospects in terms of therapeutic delivery vehicles.

Ovarian cancer (OC), cervical cancer (CC), and endometrial cancers (EC) are the three most frequent gynaecologic malignancies, and they contribute considerably to the global cancer burden. The most common gynaecological malignancy cancer-related cause of death is ovarian cancer. More than 70% of OC patients are diagnosed at an advanced stage and relapse rapidly after initial treatment; thus, the 5-year survival rate of OC is low [1,2]. With an estimated 604,000 new cases and 342,000 deaths globally in 2020, cervical cancer is the fourth most common malignancy among women. CC is a cancer that originates in cells at the junction of cervical squamous cell carcinoma, and infection with a “highly carcinogenic” strain of human papillomavirus (HPV) is a necessary but inadequate cause of CC [3]. The second most frequent cancer of the female reproductive system and the sixth most frequent cancer in women is endometrial cancer [3]. It is the most common gynaecological cancer in high-income countries, and its incidence is increasing globally [4]. Gynaecological cancer patients have a survival percentage when the disease is caught, and effective clinical treatment and improved prognosis are also significant. Sensitive, specific, and peripherally usable biomarkers are urgently needed.

Recently, significant progress was made in the study of exosomes, which represent an understudied form of intercellular communication and are important carriers for the uptake, transport, and release of cargoes such as biomarkers or therapeutic targets. The use of therapeutic exosomes to help patients is a novel and potentially promising available technology and therapy target, and the technologies already available can be leveraged to identify useful diagnostic or prognostic markers in these patients. This article reviews the research progress as well as systematic knowledge of exosomes in gynaecological malignancies in recent years, to provide a new perspective for the treatment of gynaecological malignant tumours and encourage the clinical application of exosomes in gynaecological oncology.

## 2. Structure and Function of Exosomes

The existence of extracellular vehicles (EVs) was initially hypothesized in 1946 and confirmed in 1967 [5]. EVs have been divided into distinct types based on size, particular surface characteristics, biogenesis, and content, mainly including apoptotic bodies, microvesicles, and exosomes. Among the three types of extracellular vesicles, exosomes have the highest theoretical and application value and have been the most well-researched and widely used. Therefore, extracellular vesicles are mostly referred to as exosomes, and exosomes (exosomes, exo) are often mixed with extracellular vesicles (extracellularvesicles, EVs) in academic papers and daily communication.

Exosomes are one type of EV first discovered by Bonucci and Anderson in the late 1960s [6,7]. These particles are extracellular vesicles with a cell-like structure, typically ranging in diameter from 30 to 150 nm and with a density of 1.13~1.19 g/mL, making them the smallest nanoscale EVS [8]. Exosomes are vesicles produced by endocytic cells that have a lipid bilayer membrane structure and form multivesicular bodies through intracellular invagination before fusing with the plasma membrane to be released.

Since consensus has not yet emerged on specific markers of EV subtypes, Minimal Information for Studies of Extracellular Vesicles (MISEV2018) published by the International Society of Extracellular Vesicles (ISEV) recommended the use of nomenclature indicating physical characteristics of EVs, such as size (small EVs (sEVs) < 100 nm or <200 nm; medium/large EVs (m/lEVs) > 200 nm); density (low, middle, high); or biochemical composition (e.g., CD63 + EVs) or descriptions of conditions or cellular origin (e.g., podocyte EVs) [9].

Under physiological and pathological conditions, nearly all cell types produce exosomes, which were originally thought to be waste products that eliminated undesired cellular components [10]. They have now been demonstrated to play key roles in cell-to-cell communication through cell-to-cell transfer of nucleic acids as well as specific repertoires of proteins and lipids [11].

Genetic material such as DNA, coding or noncoding RNA, proteins, lipids, and metabolites are among the various components found in exosomes [12]. Exosomes might vary in size and cargo even if they originate from the same cell (Figure 1). Exosomes have recently been revealed to play important roles in many physiological and pathological processes, including cell proliferation, apoptosis, angiogenesis, inflammatory pathways, tumour pathogenesis, and tissue damage and repair. Exosomes can be utilized as biomarkers for illness diagnosis because they are released by many cells and have various components and functions. The emerging liquid biopsy is the application of exosomal microRNAs (miRNAs) as novel diagnostic and prognostic biomarkers [13].

The lipid bilayer membrane structure of exosomes can effectively safeguard the chemicals they contain. Engineered exosomes attach cell- or tissue-targeting peptides to the surface of exosomes in order to achieve selective targeting to specific cells or tissues, in order to modulate the function of target cells as well as living tissues. Thus, they could be utilized as carriers for drug delivery, targeting specific cells or tissues in order to improve therapeutic efficacy and safety [14].

## 3. Exosomes and Gynaecologic Malignancies

Exosomes play a crucial role in intercellular communication [15], Tumour-derived exosomes have emerged as mediators of tumour formation and progression, metastatic spread, enhanced angiogenesis, and drug resistance, by regulating stromal cells and the tumour microenvironment (TME). Tumour exosomes primarily contain three major components, namely DNA, RNA, and protein [16], and their biogenesis, mechanisms involved in tumorigenesis, development, and treatment, as well as biomarker development in cancers are under investigation [17,18].

### 3.1. Exosomes in Ovarian Cancer

One of the top causes of death for women is ovarian cancer. More than 70% of OC patients are identified at an advanced stage, where the 5-year overall survival rate is lower, and this stage has an average 5-year survival rate of about 47% for all OC patients [2]. OC is the most common cause of death from gynaecological malignancies globally. OC patients are usually diagnosed at an advanced stage, in part because of the lack of early diagnostic tools, and OC patients are often prone to rapid relapse, so focused therapy needs to be studied more, along with the response detection of OC and developing more applications, which is crucial for reducing OC mortality.

#### 3.1.1. Exosomes Derived from Body Fluids

Exosomes use for early diagnosis and effective treatment of OC has advanced significantly in recent years. Nucleic acids and proteins identified in the serum of OC patients are helpful for the early diagnosis of OC. These chemicals can be utilized as diagnostic or prognostic indicators because it has been established that they are connected to the ovary. Zhu et al. [19] discovered that the expression level of miR-205 in plasma exosomes of OC patients was significantly higher as compared to that of benign and control groups. In addition, the level of miR-205 in serum of OC patients with stage III-IV was higher than that of I-II. FIGO stage III/IV, high grade, ascites, higher levels of CA-125, lymph node metastasis, and prognosis were strongly correlated with low plasma exosome-derived fragile site-associated tumour suppressor (FATS) levels in patients with OC [20]. Xiong et al. [21] explored that miR-200b was increased in serum exosomes of OC patients, and inhibited KLF6 expression in order to promote macrophage M2 polarization in OC to play a cancer-promoting role. Seven miRNAs were found to be upregulated and two miRNAs to be downregulated in the serum exosomes of OC patients, according to an exosomal miRNA study. Further analysis revealed that miR-4732-5p may be a promising candidate biomarker for the diagnosis of epithelial OC [22].

#### 3.1.2. Exosomes Derived from Cells

Exosomes from OC cells have recently been found to include a range of chemicals that have been shown to be associated with tumour progression, metastasis, angiogenesis, or drug resistance. OC cell-derived exosomes induce premetastatic niche formation, laying the groundwork for rapid metastatic invasion in a distant TME [23]. For instance, exosomal ANXA2 from OC cells promotes the mesothelial–mesenchymal transition (MTT) as well as the degradation of the extracellular matrix of human peritoneal mesothelial cells. This ultimately influences the premetastatic microenvironment of OC, providing conditions for the intraperitoneal implantation and metastasis of OC [24]. Thus, tumour-derived exosomes could serve as biomarkers suitable for liquid biopsy and new roles as chemotherapeutic targets.

MiR-21-5p [25], long noncoding RNA (lncRNA) SOX2-OT [26], and circular RNA (circRNA) Foxo3 [27] were from exosomes derived from OC cells, increasing their malignant phenotype and promoting ovarian progression. Exosomal miR-155-5p reprograms macrophages to suppress antitumour immune responses [28]; exosomal CD47 improves macrophage phagocytosis and prevents peritoneal spread by increasing phagocytosis [29]. Moreover, exosomal miRNA-29a-3p secreted by tumour-associated macrophages (TAMs) could promote OC cell proliferation and immune escape [30]; plasma cell-derived exosomal miR-330-3p induces a mesenchymal phenotype in OC as well as promotes tumour growth [31].

Exosomal circRNA 051239 [32] and CD44 [33] derived from high-metastatic OC cells could promote the metastasis of OC cells; by releasing miR-6780b-5p to support OC cells’ epithelial–mesenchymal transition (EMT), ascites-derived exosomes encourage metastasis [34]. Proteins carried by exosomes also play a significant part in OC metastasis [35].

Ovarian cancer cell-derived exosomes encourage the angiogenesis and migration of vascular endothelial cells in vitro and in vivo, which includes exosomal miR-130a [36], lncRNA ATB [37], and PKR1 [38] playing a role in it, while miR-92b-3p [39] secreted by OC cells is antiangiogenic in OC.

Drug-resistant OC cells’ exosomal miR-429 [40] and miR-21-5p [41] can confer chemoresistance on other OC cells. Reduced O-GlcNAcylation of SNAP-23 promotes exosome release in OC cells, which enhances exosome-mediated efflux of cisplatin from cancer cells, which leads to increased chemoresistance [42]. Alharbi et al. explored that the degree of platinum resistance induced in OC cells differed when exposed to low oxygen tension (1% oxygen), so they identified a set of glycolysis-related proteins and it was illustrated that chemoresistance transmission to OC cells by exosomes is related to hypoxia-induced glycolysis pathway protein expression [43].

In platinum-resistant OC cell lines, TMEM205 transmembrane protein expression is 10–20 times higher, which may contribute to OC through exosome-mediated platinum drug efflux [44]. Exosomal CLPTM1L from a cisplatin-resistant OC cell line is capable of conferring cisplatin resistance in a drug-sensitive OC cell line [45]. Exosomes secreted by chemoresistant OC cells could promote angiogenesis, miR-130a in exosomes might play a crucial role in this process [36], and targeted delivery of exosomal miR-484 could induce normalization of blood vessels, sensitizing OC to chemotherapy [46].

### 3.2. Exosomes in Cervical Cancer

Cervical cancer is the fourth most common cancer in women globally, and despite being among the most preventable cancers, CC is consistently the second leading cause of cancer death among women between the ages of 20 and 39 [2]. CC develops in the squamocolumnar junction of the cervix, and the human papillomavirus is thought to be responsible in the great majority of cases (HPV). Human papillomavirus (HPV) appears to be a major cause of cervical squamous cell carcinoma and has been the focus of research on CC diagnosis as well as treatment over the past few decades [47]. Exosomes also play a crucial part in the growth of CC. The progression of CC initially occurs in the form of local expansion, so the creation as well as maintenance of a TME, which supports the growth and spread of tumour cells, is the key to CC progression. An integral role is played by CC cell-derived exosomes in intracellular communication, which promotes tumour growth.

#### 3.2.1. Exosomes Derived from Body Fluids

Several nuclear transporters in exosomes of CC cells were identified in various studies and their presence was also verified in serum, combined as a set of biomarkers, and identified as potential biomarkers for diagnosis [48]. The level of serum exosomal lncRNA DLX6-AS1 in CC patients was significantly higher, as compared to that in CIN patients and normal controls [49]. In contrast to healthy controls, the plasma exosomal miR-125a-5p expression level of CC patients was significantly lower, which might be a potential marker to distinguish noncervical cancer and cervical cancer [50].

#### 3.2.2. Exosomes Derived from Cells

The progression of CC largely relies on tumour angiogenesis, which is dependent on tight interactions among different cellular components of the TME, specifically among tumour cells, endothelial cells, and immune cells [51]. Exosomes play a crucial part in intracellular communication and interactions. Thus, exosomes have also been used for studying the underlying mechanisms of CC tumour angiogenesis.

Angiopoietins, which regulate vascular development and are essential for vascular remodelling in inflammatory conditions and tumour angiogenesis, are regulated by a receptor known as tyrosine kinase with immunoglobulin and epidermal growth factor homology 2 (TIE2). Duet et al. [52] discovered that CC cell-derived exosomes deliver TIE2 protein to macrophages, which induces the formation of TIE2-expressing macrophages (TEMs) to promote CC angiogenesis. By upregulating hedgehog–GLI signalling, CC exosomes also encourage angiogenesis in human umbilical vein endothelial cells (HUVECs), and exosomes from HPV-positive (SiHa and HeLa) cells are more angiogenic [53]. miR-663b is also confirmed to exist in CC exosomes and promote angiogenesis by inhibiting the expression of vinculin in vascular endothelial cells [54].

MiRNAs, lncRNAs, and other functional RNAs can be transported between cells by exosomes. Among them, exosomal miR-1323 was secreted by cancer-associated fibroblasts (CAFs) transferred to CC cells, which promote CC progression and radioresistance [55]; while exosomal miR-1468-5p released by CC cells increases tumour immune escape through immunosuppressive effects by lymphatic endothelial cells (LECs) in TME, high serum exosomal miR-1468-5p levels correlate with an immunosuppressive state and poor prognosis in CC patients [56]. Similarly, exosomal miR-142-5p secreted by CC cells also mediates immunosuppression by inhibiting indoleamine 2, 3-dioxygenase expression by LECs [57]. Exosomal miR-663b promotes EMT and metastasis in CC cells by targeting MGAT3 under TGF-β1 stimulation [58]. LncRNAs have also been found in exosomes, and exosomal lncRNA UCA1 from CC stem cells promotes self-renewal and differentiation of CC stem cells through the miRNA-122-5p/SOX2 axis [59]; exosomes from cancer cells produce lncRNA AGAP2-AS1, which regulates the miR-3064-5p/SIRT1 axis to boost the proliferation of CC cells [60]; LncRNA LINC01305 can also be transferred to recipient cells via exosomes in order to enhance CC progression [61].

Exosome-carrying proteins are also implicated in the development of CC. For instance, Wnt2B protein from CC cells is delivered to fibroblasts in exosome form, where it induces fibroblast activation into CAFs and advances CC [62]. HPV E6 transcripts were also detected in exosomes of CC cells, which might serve as potential exosome biomarkers for CC [63]. From these instances, it could be recognized that exosomes play a huge role in supporting CC progression. In addition, the clinical value of exosomes in CC diagnosis and treatment is worth exploring.

### 3.3. Exosomes in Endometrial Cancer

The second most frequent cancer of the female reproductive system and the sixth most frequent cancer in women is endometrial cancer [3]. EC originates in the lining of the uterus and occurs primarily in postmenopausal women. Exosomes are a key pathway utilized by tumour cells in order to establish a supportive microenvironment.

#### 3.3.1. Exosomes Derived from Body Fluids

In endometrial liquid biopsy, exosomes have great application prospects, including nucleic acids carried by exosomes isolated from peritoneal fluid, urine, and serum of EC patients that may become new diagnostic biomarkers for EC.

Plasma-derived exosomal miR-15a-5p and exosomal lectin galactoside-binding soluble 3 binding protein (LGELS3BP) in EC patients were significantly elevated compared with controls, among which the integration of miR-15a-5p and serum tumour markers (CEA and CA125) achieved an AUC value of 0.899 [64]. Exosomal LGELS3BP also promotes EC cell growth and HUVEC angiogenesis [65]. Fan et al. also screened some miRNA markers in EC patient serum and verified the consistency in EC serum or plasma exosomes; exosomal miR-20b-5p [66] and miR-151a-5p [67] were considered as potential noninvasive biomarkers for EC diagnosis.

#### 3.3.2. Exosomes Derived from Cells

Exosomes could transport functional RNAs between cells. miR-192-5p released by TAMs, miRNA-503-3p secreted by human umbilical cord blood mesenchymal stem cells (hUMSCs), and miR-765 derived from CD8+ T cells can all be transferred into EC cells and inhibit EC progression [68,69,70]. Along with miRNAs, exosomal lncRNA NEAT1 from CAFs downregulates the miR-26a/b-5p-mediated STAT3/YKL-40 pathway to promote EC progression [71]. M2-polarized macrophages release exosomes hsa_circ_0001610 for transfer to EC cells, which significantly downregulates the radiosensitivity of EC cells through endogenous competition for miR-139-5p [72]. Nevertheless, miR-26a-5p derived from EC cells could significantly decrease the migration and tube formation ability of human LECs, and could inhibit the proliferation, migration, and invasion of EC cells [73].

Exosomes have been regarded as the key components of communication between cancer cells and other cells in the TME, and RNAs are currently being investigated as significant cargoes (Table 1). Therefore, exosomes derived from other cells can regulate proliferation, migration, and other phenotypes of tumour cells. The effects on other cells, which include the promotion of angiogenesis or lymphangiogenesis, the effect on the distribution of immune cells, and the mutual communication between tumour cells, are all based on the function of exosomes, so exosomes play a role in gynaecological tumours. The mechanism of action still must be further explored, and the translation to clinical application has broad prospects, which provides a direction for the diagnosis of disease progression and the development of therapeutic targets (Appendix A).

## 4. Clinical Diagnosis and Therapeutic Applications of Exosomes in Gynaecologic Malignancies

Exosomes play an important role in the diagnosis and prognosis of illnesses, which places great demands on the sensitivity and specificity of markers to detect diseases. Simultaneously, the type and quality of the tested samples also influences the biological understanding of exosomes and the development of biomarkers. Due to improvements in laboratory methods and technology, exosomes are now available for clinical use. These new therapeutic and diagnostic approaches utilizing exosomes have made some progress in the diagnosis and therapeutic applications of gynaecological malignancies.

To date, the lack of an elective procedure to separate specific extracellular vesicles populations in body fluids or abundantly released by tumour cells impedes the use of exosomes in clinic. A small subset of exosome subtypes with specific or prominent functions are masked by a large number of nonfunctional EVs. In fact, the available technical procedures do allow for the distinction of EVs based on their size and density, regardless of endosomal or plasma membrane origin. Therefore, there is an urgent need to develop techniques that would the isolation of a pure exosome fraction from bulky vesicular populations, as well as to comprehensively define the many subtypes of EVs.

### 4.1. Diagnostic Technology

Exosome-involved liquid biopsy is a new noninvasive and individualized method that can provide valuable information for the diagnosis of low-access tumours through the presence of tumour substances in body fluids [75], This could address the lack of sensitivity, specificity, and survival benefit of serum markers [76], as well as the highly invasive, local sampling of tissue biopsies.

Whether serum or plasma, isolation of EVs from these two blood components holds the potential to utilize EVs as disease biomarkers. Nevertheless, it remains unclear whether distinct EV subsets exist in plasma and serum. According to the research, blood sampling methods, including the anticoagulant used and the centrifugation protocol chosen, might influence the EV analysis [77]. By combining size-exclusion chromatography (SEC) with OptiPrep density gradient centrifugation, Vergauwen et al. [78] fractionated blood plasma to obtain EVs for a deeper biological understanding of EVs and the development of biomarkers. Cho et al. [79] screened noncoding RNAs from plasma exosomes, examined the association between ncRNA–mRNA networks and cancer, and built a method to screen eight types of RNA combinations as a new method for CC diagnosis. Krishnan et al. [80] prepared a new material, Chitosan grafted butein (CSB), as well as processed CSB-modified flexible screen-printed electrodes for electrochemical biosensing of exosomal CD24-specific nucleic acids at ultralow sample concentrations, which is expected to be utilized in OC diagnosis. The fluorescent gold nanoclusters with protein templates have highly fluorescent properties and biocompatibility. Combining them with exosomes successfully obtained nuclear staining of CC cells and was compatible with membrane-staining dyes, which proposes that the use of exosome synthesis for cellular imaging applications is also feasible [81].

### 4.2. Therapeutic Advances

As natural intercellular information carriers, exosomes are one of the ideal targets for the development of drug-delivery vehicles due to their nanoscale size, excellent stability, and biocompatibility. Exosome-based drug delivery has the potential to reach cells and tissues that are currently inaccessible by other drug-delivery technologies. Exosomes also have the advantage of low toxicity and low immunogenicity, reducing adverse effects on major organ systems (especially the heart) and reducing the risk of rejection and inflammation. HEK-293T cell-derived exosomes have been loaded with safranin and curcumin compounds as chemotherapeutic agents. ExoCrocin and ExoCurcumin enter tumour cells, and the synergistic effect of HPV L1-E7 polypeptide vaccine construction could significantly induce T cells’ immune response and antitumour effects [82]. Liposome nanoparticles containing ruthenium (II)–curcumin complexes are significantly cytotoxic to Hela cells and exhibit anticancer properties [83]. Bhatta et al. [84] established a multivalent phosphatidylserine binder named ExoBlock in order to block the activity of human OC-associated immunosuppressive exosomes as well as enhance T-cell-mediated tumour-suppressive effects. The cytotoxic drug paclitaxel (PTX) improved the production of exosomes, and this exosome-mediated drug efflux attenuated drug function. Omeprazole and GW4869 were discovered to be exosome inhibitors that can stop the efflux of PTX [85].

Exosomes derived from immune cells, cancer cells, and normal cells activate tumour immunity and show great potential in tumour therapy [86]. Among immune cells, exosomes released by B cells, dendritic cells (DCs), macrophages, and plasma cells activate tumour immunity by expressing tumour antigens, functioning in antigen presentation, triggering T-cell responses, and promoting cytokine release, so they have the potential to become a carrier of tumour vaccines.

Exosomes are involved in the pathogenesis, progression, and metastasis of gynaecological malignancies. Therefore, inhibiting the release or uptake of exosomes may be an effective method to inhibit tumour progression or metastasis. The acidic TME can promote the release of exosomes [87], which may alleviate the accumulation of toxic substances in cells [88]. Therefore, proton pump inhibitors have been used to reduce exosome levels in cancer models [85,89]. Lee et al. [90] found that the knockdown of monocarboxylate transporter 1 (MCT1) and its partner CD147 reduced the release of exosomes from glioma cells, while overexpression significantly increased the release of exosomes. It is suggested that MCT1 and CD147 may play a key role in suppressing exosome secretion in tumours. In addition, various exosome uptake inhibitors, including amiloride, dynasore, chlorpromazine, and heparin [91], have been developed to target the process of exosome uptake by recipient cells, which is dependent on different molecules and glycoproteins on exosomal membranes and recipient cells.

### 4.3. Overcome Chemoresistance

About 90% of cancer-related deaths are attributable to drug resistance, a significant obstacle to effective cancer treatment. The potential of exosomes to overcome cancer drug resistance has been exploited. Exosomes can bypass endosome capture and diffuse uniformly into the cytoplasmic matrix to enhance the anticancer effects of chemotherapy when used to transport cisplatin into cisplatin-resistant OC cells [92]. Tumour suppressor miRNAs are new targets for tumour therapy, but the difficulty of miRNA delivery limits its clinical application. Exosomes have been used as carriers for OC miRNA replacement therapy, and the synthesized miRNAs are loaded into exosomes by electroporation, which might provide a new direction for exosomes as vector of drug delivery to increase tumour treatment sensitivity [93]. Exosome-mediated targeted delivery of miR-484 causes vascular normalization and reconnection of tumour vasculature and makes OC cells more chemosensitive [46]. Overexpression of miR497 could overcome OC chemotherapy resistance by inhibiting the mTOR pathway. Therefore, Li et al. created an exosome–liposome hybrid nanoparticle codelivery of TP and miR497, which could effectively enrich in the tumour area, improve tumour cell apoptosis, exert significant anticancer activity, and overcome chemoresistance in OC [74]. Exosomes are anticipated to have a significant role in the treatment of drug-resistant gynaecological cancers in the future (Figure 2).

## 5. Conclusions and Prospects

Research on exosomes primarily concentrates on specific biomarkers with diagnostic and prognostic implications as well as therapeutic targets in gynaecological malignant disorders. Exosomes are mostly studied in OC due of their quick development, high recurrence rate, and poor prognosis. Evidence accumulated in the past suggests that some exosomes have strong tumour-promoting effects in gynaecological malignancies. Nevertheless, the role of tumour-derived exosomes and their cargo still needs further exploration in order to clarify their roles, mechanisms, and application prospects. In order to better understand tumour progression, current research has been focused on developing exosome-based diagnostic as well as prognostic tools for the effective control and management of gynaecological malignancies.

With noninvasive and individualized advantages in gynaecological oncology applications, exosome participation in liquid biopsies, which are characterized by the analysis of tumour material in the peripheral circulation, might provide valuable information for the diagnosis of low-access tumours. So far, the potential shown by exosomes in the early diagnosis of tumours is expected to be a promising alternative to traditional tissue-sampling methods, but is still in its infancy. More research is needed to further elucidate the mechanism of exosome release, identification of tumour tissue origin, and biological significance, and to improve technical stability and reproducibility by implementing standardized procedures for clinical application. Exosomes in serum are expected to serve as biomarkers for screening or early diagnosis of cervical cancer, which can help overcome the limitations of sampling locations and personnel due to the invasiveness of cervical cancer screening and narrow the gap between developed countries and the rest of the world.

Precision medicine was first proposed in 2015, and since then, people have been seeking accurate precise diagnosis and customized therapy. Exosomes have become a new research hotspot because of their widespread existence, stability, and ease of access in vivo, and have great prospects in assisting in accurate diagnosis and the treatment of diseases. Exosomes have benefits over other vesicles, in that they can carry genetic material or proteins for intercellular communication and material transfer. As a result, there is a growing study interest in exosomes’ role as medication carriers. The engineered exosomes, which have been constructed by modifying the surface molecules of exosomes to endow them with cell- and tissue-targeting specificity, are powerful tools in order to achieve specific cell-targeted transport. Proteins or other small molecules can have therapeutic effects on specific disease areas or cells by loading them with functioning genetic material. Exosomes may also be used as nanocarriers to deliver chemotherapeutic drugs in order to overcome drug-resistant tumours. Further investigation and demonstration and standardized clinical trials are needed to verify and ensure safety and accuracy before clinical routine use is realized.

In the future, regular monitoring with liquid biopsies to elucidate resistance acquired due to genetic alterations, such genome-based prediction of drug response, may enable liquid biopsies as a companion biomarker for large-scale drug trials [94]. In addition, tumour-derived exosomes themselves may also become targets for tumour therapy, including inhibiting the critical role of exosomes in tumour metastasis. The ability of exosomes to induce antitumour immune responses in the cancer environment can be used to develop safe and reliable tumour vaccines [95].

In conclusion, exosomes may be an important player in addressing some of the key unanswered questions in the onset, progression, and treatment of gynaecological cancers. It is hoped that more and more research will help diagnose and treat cancer and improve outcomes for gynaecological cancer patients around the world.

## Figures and Tables

**Figure 1 cancers-14-04743-f001:**
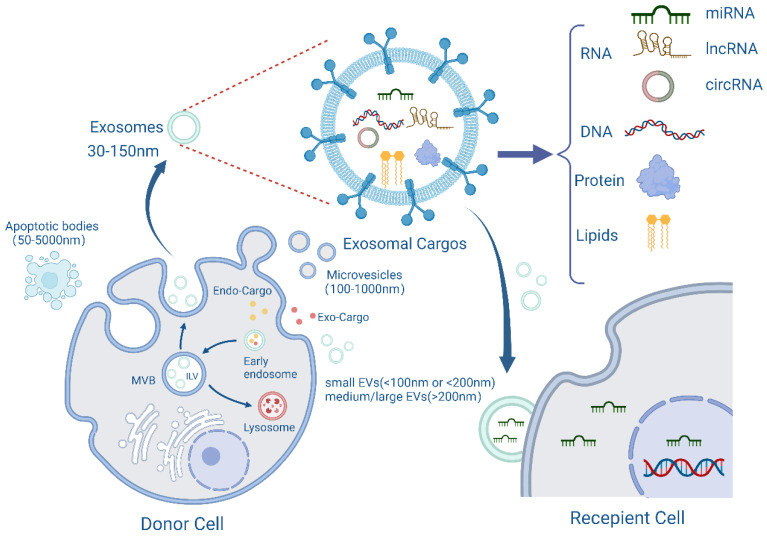
The biogenesis, content, and transport of exosomes. Exosomes originate from plasma membrane invaginations and then form early endosomes, which in turn form multivesicular bodies (MVB) containing intraluminal vesicles (ILVs). Some MVBs then fuse with the plasma membrane and release ILVs into the extracellular environment as exosomes. During the formation of exosomes, they will carry exogenous or endogenous cargoes, and finally be released from the cells with a diameter of 30–150 nm. The membrane structure and contents of exosomes, including RNA, DNA, proteins, lipids, etc., are also shown in the figure, which are then transported to recipient cells for their functions. Created with BioRender.com.

**Figure 2 cancers-14-04743-f002:**
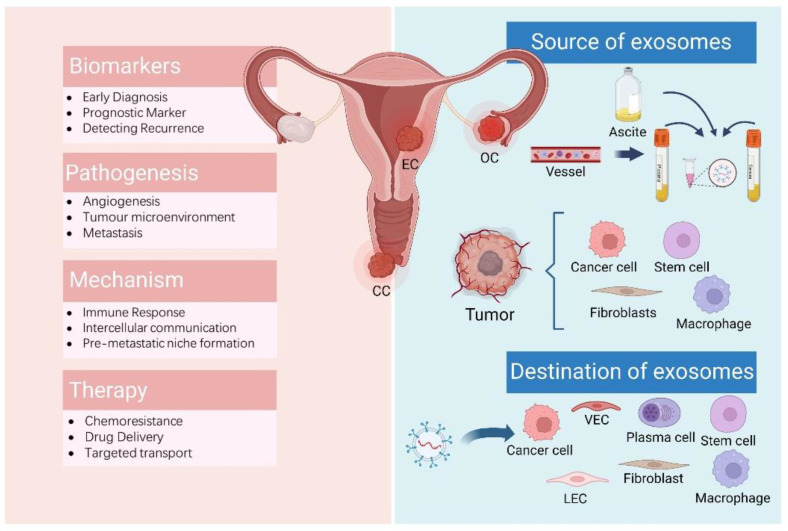
The sources and destination of exosomes. Exosomes can be derived from body fluids such as blood or ascites, and from a variety of cells, including tumour cells, immune cells, stem cells, etc., and are transported to a variety of receptor cells to play their role. Exosomes are involved in various physiological or pathological processes of diseases and are expected to be utilized as markers for disease diagnosis and prognosis, or to play a therapeutic role in clinical applications. OC: ovarian cancer; CC: cervical cancer; EC: endometrial cancer; LEC: lymphatic endothelial cells; VEC: vascular endothelial cell. Created with BioRender.com.

**Table 1 cancers-14-04743-t001:** RNAs derived from exosomes from gynaecological malignancies.

Disease	Exosomal Cargo	Type	Exosome Derivation	Recipient Cells	Clinical Value	References
OC	miRNA-205	miRNA	Serum	-	Diagnosis	[19]
	miR-200b	miRNA	Serum	Macrophage	Diagnosis and therapeutic target	[21]
	miR-4732-5p	miRNA	Serum	-	Diagnosis and monitoring progress	[22]
	miR-21-5p	miRNA	OC cells	OC cells	Progression and therapeutic target	[25]
	lncRNA SOX2-OT	lncRNA	OC cells	OC cells	Progression and therapeutic target	[26]
	circRNA Foxo3	circRNA	OC cells	OC cells	Progression	[27]
	miR-155-5p	miRNA	OC cells	Macrophages	Inhibiting progression	[28]
	miR-29a-3p	miRNA	Macrophages	OC cells	Progression	[30]
	miR-330-3p	miRNA	Plasma cells	OC cells	Therapeutic target	[31]
	circRNA051239	circRNA	OC cells	OC cells	Metastasis	[32]
	miR-6780b-5p	miRNA	Ascites	OC cells	Metastasis	[34]
	miR-130a	miRNA	OC cells	HUVECs	Angiogenesis	[36]
	lncRNA ATB	lncRNA	OC cells	HUVECs	Therapeutic target	[37]
	miR-92b-3p	miRNA	OC cells	HUVECs	Antiangiogenic therapy	[39]
	miR-429	miRNA	OC cells	OC cells	Chemoresistance and therapeutic target	[40]
	miR-21-5p	miRNA	OC cells	OC cells	Chemoresistance and therapeutic target	[41]
	miR-484	miRNA	Engineered	OC cells and AECs	Chemotherapy sensitization	[46]
	miR-497	miRNA	Engineered	OC cells	Overcoming chemoresistance	[74]
CC	lncRNA DLX6-AS1	lncRNA	Serum	-	Diagnosis	[49]
	miR-125a-5p	miRNA	Plasma	-	Diagnosis	[50]
	Hedgehog-GLI	-	CC cells	HUVECs	Angiogenesis	[53]
	miR-663b	miRNA	CC cells	HUVECs	Angiogenesis	[54]
	miR-1323	miRNA	CAFs	CC cells	Progression and therapeutic target	[55]
	miR-1468-5p	miRNA	CC cells	LECs	Prognostic markers and therapeutic target	[56]
	miR-142-5p	miRNA	CC cells	LECs	Diagnostic marker and therapeutic target	[57]
	miR-663b	miRNA	CC cells	CC cells	Metastasis	[58]
	lncRNA UCA1	lncRNA	CC stem cells	CC stem cells	Progression	[59]
	lncRNA AGAP2-AS1	lncRNA	CC cells	CC cells	Therapeutic target	[60]
	LINC01305	lncRNA	CC cells	CC cells	Progression	[61]
EC	miR-15a-5p	miRNA	Plasma	-	Diagnosis	[64]
	miR-143-3p,miR-195-5p, miR-20b-5p,miR-204-5p, miR-423-3p,miR-484	miRNA	Serum	-	Diagnosis	[66]
	miR-142-3p,miR-146a-5p,miR-151a-5p	miRNA	Plasma	-	Diagnosis	[67]
	lncRNA NEAT1	lncRNA	CAFs	EC cells	Therapeutic target	[71]
	hsa_circ_0001610	circRNA	TAMs	EC cells	Radioresistance	[72]
	miR-26a-5p	miRNA	EC cells	LECs	Metastasis	[73]

OC: ovarian cancer; CC: cervical cancer; EC: endometrial cancer; CAFs: cancer-associated fibroblasts; TAMs: tumour-associated macrophages; hUVECs: human umbilical vein endothelial cells; AECs: angiogenic endothelial cells; LECs: lymphatic endothelial cells.

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
