# Peer review of "Advances in Exosomes as Diagnostic and Therapeutic Biomarkers for Gynaecological Malignancies"

_cancers, 2022, doi:10.3390/cancers14194743_

Round 1
Reviewer 1 Report
Excellent and comprehensive review of the current applications and prospects of using Exosomes in the management of Gynaecological cancers.
I liked the way the data is presented broken down per tumour type.
Also liked the graph with the different applications of exosomes in diagnosis/treatment/resistance.
The most impressive and immediately applicable exosome technology is the quoted: 'EC patients had been significantly elevated compared with controls, among which, the integration of miR-15a-5p and serum tumour markers (CEA and CA125) was achieved AUC value of 0.899'. The AUC value is impressive and I think that this could be applied directly to clinical practice.
Author Response
Thank you verymuch for your kind comments and acknowledgment of our review. We hope that our review can provide new insights and ideas for the involvement of exosomes in the diagnosis and treatment of gynecological malignancies.
Reviewer 2 Report
For the authors
The review by Meng-Dan Miao et al. examines the potential diagnostic and therapeutic role of exosomes in several type pf gynecological malignant tumors. It offers a complete and updated overview of exosomes obtained either from body fluids or in in vitro models, thus providing insights into the definition of novel biomarkers.
The work is interesting, but it presents some flaws that hampers it to be published in the present form.
I strongly recommend to make the corrections as indicated below, and to include and discuss some specific points, that will complete and considerably improve the manuscript.
1. Pag.2 chapter 2. EVs naturally secreted from all type of cells are composed of subsets of small vesicles with different molecules composition, biogenesis and biological roles. The authors never mention the big debate that was developed by the scientific community, leading to the specific guide lines (MISEV 2018), and the position statement released by ISEV. Accordingly, major key points have to be provided before naming a vesicles population as exosomes.
I suggest to add few sentences on this important aspect, together with the exact definition on the vesicles population examined in the reported papers, if possible.
Moreover, I suggest to add in the figure 1 and briefly describe in the text all the vesicle types of the EVs population.
2. Figure 1. The figure representing intracellular vesicles biogenesis and multivesicular bodies (MVB) formation is too generic and some details regarding ESCRT dependent and independent pathway should be included, such as the formation of MVB. The release of ILVs as exosomes from MVBs should be indicated also.
3. Pag 3 chapter 3. First paragraph: “mechanism” of what? Please complete the sentence
4. I suggest to add a table listing all miRNAs, lncRNAs in exosomes derived from body fluids or cells discussed throughout the text with cell derivation and application, in order to summarise and present in a concise way the molecules described.
5. Table 1 is incomprehensible to me. Does this table summarise the molecules described in the text? If yes all the molecules discussed should be inserted in the table.
6. Please cite Table 1 somewhere in the text.
7. To date the lack of an elective procedure to separate specific extracellular vesicles populations that are abundantly released by tumour cells, impedes the use of exosomes in clinic. In fact the available technical procedures do allow the distinction of EVs based on their size and density, regardless on endosomal or plasma membrane origin. Therefore there is an urgent need to develop techniques that would ensure the isolation of a pure exosome fraction from the bulky vesicular population.
This important aspect on the use of exosome in gynecological malignancies should be discussed in the revised version.
8. Pag 10. “Its delivery separation, construction, transplantation and delivery process still required further improvement, and needs to be simplified and perfected”. This sentence is unclear and not linked to the text. I suggest to remove it or to provide explanations.
Minor points
1. Pag 7 paragraph 4. Please revise English form
2. Pag 8 paragraph 4.2. last sentence is incomprehensible
3. 4.3 please revise English form
4. Pag 9 please change “external” with circulating
Reviewer 3 Report
In the manuscript “Advances in exosomes as diagnostic and therapeutic biomarkers for gynaecological malignancies”, Miao et al discussed about the advancements in exosome based diagnostic and therapeutic approaches, however it requires following major revision before its consideration for publication.
There are several shortcomings, which need to be addressed before this manuscript can be considered for publication.
1. Authors should discuss why the exosomes have been emphasized, and how it has advantages over existing methods for diagnostic purposes.
2. Authors should discuss how exosome-based drug delivery including advanced methods such as microfluidic drug loading in exosomes can be utilized for therapeutic purposes.
3. Abstract and conclusion should be revised to encompass the entire result and constructive future outlooks as a discussion.
4. In the conclusion section, future perspectives, or outlook in the context of exosome-based clinical advancements should be discussed.
5. Following latest literatures about EVs are worth to discuss in the manuscript in the context of LPR and AFM diagnostic approaches via non-invasive detection of exosomal cargoes:
Site specific biotinylated antibody functionalized Ag@AuNIs LSPR biosensor for the ultrasensitive detection of exosomal MCT4, a glioblastoma progression biomarker.
Label-free sensing of exosomal MCT1 and CD147 for tracking metabolic reprogramming and malignant progression in glioma
Exosomes: Small vesicles with big roles in cancer, vaccine development, and therapeutics.
Round 2
Reviewer 3 Report
Satisfactory revision.